# Mechanism for Reducing the Horizontal Transfer Risk of the Airborne Antibiotic-Resistant Genes of *Escherichia coli* Species through Microwave or UV Irradiation

**DOI:** 10.3390/ijerph19074332

**Published:** 2022-04-04

**Authors:** Azhar Ali Laghari, Liming Liu, Dildar Hussain Kalhoro, Hong Chen, Can Wang

**Affiliations:** 1School of Environmental Science and Engineering, Tianjin University, Tianjin 300350, China; azharlaghari18@hotmail.com (A.A.L.); liuliming_1998@163.com (L.L.); 2Department of Veterinary Microbiology, Faculty of Animal Husbandry and Veterinary Sciences, Sindh Agriculture University Tandojam, Hyderabad 70050, Pakistan; drdildarkalhoro@gmail.com; 3Tianjin Key Laboratory of Indoor Air Environmental Quality Control, Tianjin 300350, China

**Keywords:** antibiotic-resistant bacteria (ARB), antibiotics-resistant genes (ARGs), microwave, UV, horizontal gene transfer (HGT)

## Abstract

Antibiotic-resistant bacteria (ARBs) and antibiotic-resistant genes (ARGs) as new types of contaminants are discharged into the environment, increasing the risk of horizontal gene transfer (HGT). However, few researchers have examined the impacts of airborne ARB deactivation on HGT risk. The deactivation of airborne *Escherichia coli* 10667 (carrying *sul* genes) and the emission and removal of ARGs were mainly investigated in this study. Moreover, the potential mechanisms of HGT and transfer frequencies under microwave (MW) and ultraviolet (UV) irradiation were investigated using the nonresistant *E. coli* GMCC 13373 and *E. coli* DH5α with plasmid RP4 as the recipient and donor, respectively. *E. coli* CICC 10667 and *E. coli* DH5α with RP4 plasmid achieve log inactivation values as high as 5.5-log and 5.0-log, respectively, which were quite different from the antibiotic-sensitive strain *E. coli* CGMCC 13373 (3.4-log) subjected to MW irradiation. For UV disinfection, *E. coli* DH5α with the RP4 plasmid was reduced at 4.4-log, *E. coli* CGMCC 13373 was reduced at 2.3-log, and *E. coli* CICC 10667 was inactivated at 2.1-log. The removal rates of ARGs and HGT frequencies under MW irradiation were compared with those under UV irradiation. The ARGs removal efficiency (85.5%) obtained by MW was higher than that obtained by UV (48.2%). Consequently, the HGT frequency (0.008) of airborne ARGs released to the recipient (forward transfer) decreased and was lower than that under UV irradiation (0.014). Moreover, the plasmid RP4 was transferred from the donor to the surviving damaged *E. coli* 10667 as cell permeability (reverse transfer) was increased at a high HGT frequency (0.003) by MW, which was close to the value by UV (0.002). Additionally, *sul1* and *sul2* genes were confirmed to be more resistant to MW than the *sul3* gene. These findings reveal the mechanism of HGT between damaged *E. coli* 10667 and surrounding environmental microbes. Microwave is a promising technology for disinfecting airborne microbes and preventing the spread of antibiotic resistance.

## 1. Introduction

As a global crisis affecting public health and the ecological environment, antibiotic resistance (AR) is caused by the excessive use and abuse of antibiotics in the field of medicine, aquaculture, and livestock farming [1,2,3]. Pathogens in the hospital environment are often resistant, presenting a significant risk to human health [4,5]. In recent years, antibiotic-resistant bacteria (ARBs) and antibiotic-resistant genes (ARGs) have been detected extensively in surface water, severely threatening the safety of drinking water [6,7]. AR is expected to cause as many as 10 million deaths annually by 2050 if no action is implemented immediately [8].

The degradation of total DNA (tDNA) containing intracellular DNA (iDNA) as well as extracellular DNA (eDNA) is highlighted by increased ARG levels. eDNA is obtained due to the loss of integrity of bacteria and effective emission from alive bacteria, whereas iDNA detects a fraction of bacteria [8,9,10]. When eDNA that benefits ARGs is secreted into the environment, other bacterial recipients utilize the naked eDNA through horizontal gene transfer (HGT), and thus ARGs transmission occurs in the same family or various families (cross-species) of bacteria [11,12,13,14]. Additionally, some damaged bacteria have a high cell membrane permeability after disinfection and quickly assimilate discharged ARGs or plasmids from sensitive donors and then become ARBs. However, a limited number of studies have determined HGT during ARB disinfection, and no study has compared microwave (MW) and ultraviolet (UV) airborne disinfection technologies with HGT. Therefore, practical and promising technologies are urgently needed to decrease ARG enrichment and inactivate ARBs.

Nevertheless, some studies have described the regrowth of surviving cells under UV irradiation [15,16] and indicated that UV-inactivated ARBs are susceptible to photo-activation due to a lack of activity in continuous disinfection [17,18]. UV and ozone treatments and chlorination are common disinfection methods. Although chlorine and ozone treatment are effective in inactivating bacteria, they often generate toxic by-products, and thus their applications are limited [19,20]. By contrast, UV is widely used in water and air disinfection.

MW irradiation can quickly denature enzymes, proteins, and membranes by disrupting cellular metabolic activity and is an effective disinfection method without undesirable disinfection by-products [19,20]. Currently, MW irradiation has been chiefly used for the disinfection of water [21], solids [22], and food [23]. However, the used of the technology in airborne disinfection is rarely reported due to the weak absorption of MW energy by air [24].

Given these challenges, one effective solution is applying MW-absorbing materials (MAMs) that effectively utilize MW energy [25,26]. Two main factors, namely interface impedance matching and electromagnetic wave loss, usually affect the microwave absorption properties of materials [25,26]. Magnetic applications based on iron materials, such as Fe, γ-Fe_2_O_3_, and Fe_3_O_4_, have the advantage of exhibiting good electromagnetic wave absorption [27]. A significant feature of iron-based materials is that they absorb radar waves through their extensive magnetic anisotropy and high magnetic susceptibility [28].

In this study, MW treatment based on absorption materials (MAMs) was compared with UV irradiation in terms of performance in inactivating ARBs and ARGs. The electrical energy per order (EE/O) performance of MW and UV for the release and degradation of ARGs was investigated. For the first time, we investigated the airborne HGT frequencies of *Escherichia coli* 10667 (containing *sul* genes) ARGs released into a recipient in the environment (forward transfer) and the survivability of injured *E. coli* 10667 (*sul*) delivered by plasmid RP4 from the environment (reverse transfer) after MW or UV irradiation.

## 2. Materials and Methods

### 2.1. Chemicals and Reagents

All chemical reagents were of analytical grade or higher purity and were purchased from various commercial suppliers. Sodium chloride (NaCl) and anhydrous sodium sulfate (Na_2_SO_4_) were purchased from Macklin (Shanghai, China). Sodium dihydrogen phosphate (NaH_2_PO_4_) and trisodium phosphate (Na_3_PO_4_) were purchased from Rhawn (China). Nutrient broth agar was purchased from Aobox Biotechnology Co., Ltd. (Beijing, China). Phosphate buffer saline, sulfanilamide, and tetracycline were purchased from Sigma-Aldrich Co., LLC (St. Loius, MO, USA). *E. coli* (CICC 10667) and *E. coli* (GMCC 13373) were purchased from the China Center of Industrial Culture Collection and the Institute of Microbiology, Chinese Academy of Sciences, respectively. *E. coli* DH5α was purchased from the Query Network for Microbial Species of China.

### 2.2. Preparation of Antibiotic-Resistant Bacterial Samples

*E. coli* (CICC 10667), which indicates higher resistance (>50 mg/L) to sulfanilamide (*sul1, sul2,* and *sul3*), was designated as the target antibiotic-resistant bacteria. *E. coli* strains are the common type of Gram-negative microorganism that can proliferate under simple culture conditions, have fast and robust propagation abilities, and are easy to control. The sequences of primers are listed in Appendix A. *E. coli* glycerol samples that were preserved at −80 °C and incubated at 37 °C in a nutrient broth medium (10 g/L peptones, 3 g/L beef extract, 5 g/L NaCl, pH 7.2) in a shaking incubator at 160 rpm for 16–24 h till a stationary phase of growth was reached [29]. The interactions among treated ARBs, environmental bacteria, *E. coli* DH5α with RP4 plasmid containing tetracycline-resistant genes (>16 mg/L), and nonresistant *E. coli* GMCC 13373 (sensitive to antibiotics) were investigated. The microbial strains were centrifuged at (10,000 rpm) for 10 min at 4 °C. The bacterial cells were triplicate washed with a sterilized saline solution (NaCl, 0.9%) and resuspended in purified double-distilled water to obtain a bacterial cell suspension with a concentration of around 10^8^ (CFU/m^3^).

### 2.3. Experimental Setup of MW and UV Irradiation Device

Figure 1a shows an aerosol generator (ATM226, Topas, Dresden, Germany) atomizing an aqueous suspension containing airborne bacteria. The airborne *E. coli* strain that comprises tetracycline-resistant genes was mixed with disinfected air passed through the MW setup from the inlet to the outlet along the MAMs. The cylinder-shaped tube in the microwave device had a diameter of 100 × 550 mm and included two MW absorption materials (Fe_3_O_4_@SiCcfs). The MW was applied at a power of 150, 300, and 500 W and irradiation time of 20 s. After microwave irradiation, an impinger sampler (AGI-30 sampler, Jolyc Technology Co., Ltd., Beijing, China) was used for obtaining emission airborne samples. The flow rate of the sampling pump was 12.5 L/min, and each sampling duration was 10 min. The inlet flow rate was the sum of F1 and F2, which was an equal flow rate to the outlet. F1 and F2 indicated the flow rate of the air pump (7.5 L/min) and the aerosol generator (5 L/min), respectively. The volume of the designated sample solution was 30 mL of 0.9% NaCl solution [30]. All collected samples were stored in sealed sterile flasks at 4 °C. Then, the samples were collected within 2 days for DNA extraction or subsequent use [31,32].

The irradiation time in this study was equal to the empty bed residence time in the reactors and was calculated using Equation (1).
(1)T=V/Q
where *T* is the irradiation time, *V* is the reactor volume, and *Q* is the flow rate. Time was controlled with a microwave switch and monitored with a flow meter.

Figure 1b is a schematic illustration of the experimental setup for inactivating airborne microorganisms. The UV device was cylindrical with a dimension of 7 cm × 47 cm and was equipped with a vertically positioned UV light. UVC irradiation (254 nm) was obtained using various power levels (e.g., 8, 14, 23, and 28 W). The bacteria containing aqueous suspension was nebulized with sterile air production of bacterial bioaerosol. The airstream containing airborne bacteria was collected and tested after passing through the UV unit from the bottom to the top. The bacteria’s exposure time to UV irradiation was adjusted by regulating the airflow rate through the UV unit between 0.26 and 0.65 m^3^/h [33]. The bacterial concentration was estimated according to the number of CFUs acquired through plate counting and divided by the volume of bioaerosol collected (m^3^). Each experiment was performed in triplicate.

### 2.4. DNA Extraction and ARGs Analysis

According to the instructions of the FastDNA Spin kit manufacturer, 5 mL of samples were pre-investigated and disinfected using a DNeasy Power water kit (Qiagen GmbH, Hilden, Germany). Then, tDNA was obtained from the filter membrane after 5 mL of sample was screened through a 0.22-m filter membrane (MP Biomedicals, CA). More detailed information can be found in the Appendix A [28,34,35]. A Q5000 micro-UV spectrophotometer was used to determine the DNA concentration (Quawell, USA). A Bio-Rad iQ5 qualitative polymerase chain reactor (qPCR, Bio-Rad Company, CA, USA) was used for analyzing the copy number of tARGs and iARGs (*sul1*, *sul2*, and *sul3*). Extracted DNA was preserved at the temperature of (−20 °C). The qPCR reaction mixtures (total amount = 25 µL) contained 9.5 µL double-distilled water (ddH_2_O) (9.5 µL), 1.0 µL forward primer and reverse primer (1.0 µL), TB Green Premix Ex *Taq* II (12.5 µL, Takara), and 1.0 µL samples (a control sample of ddH_2_O was used). The qPCR reaction was set at 95 °C for 30 s, following 40 cycles of 95 °C for 5 s and 30 s at the annealing temperature. All of the reactions were performed in triplicate.

### 2.5. Horizontal Gene Transfer Experiments

We analyzed the risks of HGT releasing ARGs into the environment by MW and UV treatment as shown in Figure 2. The *E. coli* DH5α strain with the RP4 plasmid at a concentration of 10^8^ CFU/m^3^, which encrypts a high level of tetracycline resistance (*>*16 mg/L), and the *E. coli* GMCC 13373 (non-resistant bacteria) at a concentration of 10^9^ CFU/m^3^ were used to investigate the contact between the *E. coli* 10,667 (*sul*) by a deactivation procedure (the injured *E. coli* 10,667 [*sul*]) and surrounding bacteria. Experiment (1) depicts the “Forward transfer” impact of the injured *E. coli* 10,667 (*sul)* on environmental bacteria. In contrast, experiment (2) was designated “reverse transfer” and demonstrated the influence of surrounding bacteria on injured *E. coli* 10667 (*sul*).

By contrast, experiment (2) was designated as “reverse transfer”. This experiment demonstrated the influence of surrounding bacteria on damaged *E. coli* 10667 (*sul*). 

A total amount of 5 mL treated *E. coli* CICC 10667 (*sul*) suspension was cultured in 100 mL of nutrient broth medium with the suspensions of *E. coli* GMCC 13373 (5 mL) and 5 mL *E. coli* DH5α at 37 °C separately. The mixed solution was incubated at 160 rpm for 6–48 h till a phase of maturation was reached. As a control, the non-treated *E. coli* CICC 10,667 (*sul*) suspension was utilized. Furthermore, the washed sample and placed on the Petri dish of nutrient agar with suitable antibiotics. On the left-hand side, the recipient strain (*E. coli* GMCC 13373) was investigated using free antibiotics plates. The donor and the sum of the donor and transconjugants were displayed, and plates with 50 mg/L sulfanilamide were used. On the right-hand side, the donor strain (*E. coli* DH5α), the recipient, was analyzed on plates comprising 50 mg/L sulfanilamide. The transconjugant was displayed on plates comprising sulfanilamide (50 mg/L) and tetracycline (16 mg/L). The risks of HGT were examined under MW and UV irradiation. All experiments were conducted in three replicates.

Equations (2) and (3) were used in determining the frequency of horizontal transfer for forward and reverse transfer [36].
(2)$$\mathrm{Frequency}\;\mathrm{of}\;\mathrm{horizontal}\;\mathrm{transfer}\;\left(\mathrm{forward}\;\mathrm{transfer}\right)=\frac{\left(C_{P2}-C_{P1}\right)\left(\mathrm{CFU}/m^{3}\;\right)}{C_{P3}\left(\mathrm{CFU}/m^{3}\right)}$$
(3)$$\mathrm{Frequency}\;\mathrm{of}\;\mathrm{horizontal}\;\mathrm{transfer}\;\left(\mathrm{reverse}\;\mathrm{transfer}\right)=\frac{C_{P5}\left(\mathrm{CFU}/m^{3}\right)}{C_{P4}\left(\mathrm{CFU}/m^{3}\right)}$$
where *CP1*, *CP2*, *CP3*, *CP4*, and *CP5* are the concentrations of different samples on the various nutrient agar plates.

### 2.6. Statistical Analysis of Airborne Samples

The removal efficiency of airborne microorganisms was calculated using analysis of variance and the paired *t*-test (sigma plot 10 component) at 10 min sampling time at different MW power outputs. *p* values of less than 0.05 indicated statistically enormous variation at the 95% self-assurance level.

## 3. Results and Discussion

### 3.1. Comparison of Inactivation Performance of Antibiotic Resistant and Antibiotic Sensitive Bacteria under MW or UV Exposure

This study used three types of *E. coli* strains for testing, as shown in Figure 3. The concentration of *E. coli* CICC 10667 (2.5 × 10^3^ CFU/m^3^) under MW irradiation was lower than the concentrations of *E. coli* CGMCC 13373 (3.3 × 10^4^ CFU/m^3^) and *E. coli* DH5α with RP4 plasmid (5.3 × 10^3^ CFU/m^3^) at 750 W (Figure 3a). The logarithmic inactivation efficiency of *E. coli* CICC 10667 (0.9-log. 3.3-log, 4.0-log, and 5.5-log), *E. coli* CGMCC 13373 (1.0-log, 1.2-log, 2.4-log, and 3.4-log), and (0.8-log, 2.7-log, 3.8-log, 5.0-log) was calculated under MW irradiation at 150, 300, 500, and 700 W, as shown in Figure 3c. A previous study discovered that by less than 2 min of residence time, the removal efficiency rates were 90% at 700 W, 65% at 385 W, and 50% at 119 W [37]. This study presented higher inactivation performance than the results of the previous study [14] because of the absorbing materials used.

Similarly, the survival concentration decreased when UV power increased. The concentration of *E. coli* DH5α with RP4 plasmid (2.0 × 10^3^) was lower than the concentrations of *E. coli* GMCC13373 and *E. coli* CICC 10667, which were 1.0 × 10^5^ and 5.0 × 10^4^ CFU/m^3^ at 28 W. The log inactivation efficiency was calculated for various stains under different UV irradiation power outputs, as shown in Figure 3b,d. The inactivation rates of the two *E. coli* strains with resistance to multiple antibiotics were very similar but different from those of antibiotic-sensitive strains, as observed in Figure 3c. For the model fitting of airborne microbe inactivation by MW exposure, the antibiotic-sensitive strain *E. coli* CGMCC 13373 showed better self-protection than multidrug-resistant bacteria. The inactivation efficiency of *E. coli* DH5α with the RP4 plasmid was 4.4-log, which was higher than that of *E. coli CGMCC* 13373. The inactivation efficiency rates of *E. coli* CICC 10667 and *E. coli* CGMCC 13373 were 2.1-log and 2.3-log, respectively, which were higher than the inactivation efficiency of *E. coli* CICC 10667 (Figure 3d). This result implied that different bacteria had different levels of tolerance to UV light [38]. *E. coli* DH5α with the RP4 plasmid was more sensitive to degradation during UV treatment, whereas *E. coli* CICC 10667 was more easily degraded by MW than the other strains.

### 3.2. Effect of MW or UV Irradiation on the Forward and Reverse Transfer Frequencies

Different disinfection technologies, including MW irradiation based on absorption material or UV irradiation, were used in comparing the forward and reverse transfer frequencies under various power outputs. As shown in Figure 4a, the forward transfer frequency (ARGs released from damaged *E. coli* 10667 (*sul*) into the environmental recipient) gradually reduced under MW. However, the reverse transfer frequency of plasmid RP4 from environmental bacteria to surviving damaged *E. coli* 10667 (*sul*) under MW gradually increased from 1.30 × 10^−2^ to 1.40 × 10^−2^ and even to 1.50 × 10^−2^ as power increased (0–300 W). The frequency decreased to 6.7 × 10^−3^ and 3.0 × 10^−3^ at 500 and 700 W, respectively. Figure 4b shows that the forward frequency was 1.75 × 10^−2^, 1.65 × 10^−2^, 1.50 × 10^−2^, and 1.40 × 10^−3^ under UV irradiation at 8, 16, 23, and 28 W, respectively. However, reverse transfer frequency was significantly affected by UV light. The frequencies were 1.30 × 10^−2^, 1.0 × 10^−2^, 5.0 × 10^−3^, and 2.0 × 10^−3^ under UV irradiation at the same power output.

In this study, MW significantly decreased the HGT frequencies of the emitted ARGs into the recipient (forward transfer) as compared with UV. By contrast, the reverse transfer frequency was higher than that under UV because the donor plasmid RP4 was used in transferring the surviving injured *E. coli* 10667 (*sul*) cells with increased cell permeability.

### 3.3. Potential Assessment of ARGs Horizon Transfer

The inactivation performance of ARGs and HGT frequency under MW or UV irradiation were compared. As shown in Figure 5a, the inactivation of *E. coli* 10667 (*sul*) and the removal of tARGs significantly varied between MW and UV irradiation. The inactivation rates of *E. coli* 10667 (*sul*) by MW and UV were 6.3-log and 2.5-log, respectively. MW presented a significantly higher removal efficiency (85.5%) than UV disinfection (48.2%).

The removal efficiency of tARGs under MW irradiation was considerably higher than that under UV irradiation. Exposed ARB (*sul*) showed higher resistance, and tARGs were not easily degraded by UV irradiation [39]. Furthermore, UV irradiation developed a pyrimidine dimer, restricting the iDNA copies and regulating gene expression [40,41]. As a result, the fraction of iARGs under MW irradiation was higher than that under UV irradiation. By contrast, the fraction of eARGs under MW irradiation was lower than that under UV irradiation as shown in the Appendix A in Appendix A(c) [42].

Additionally, the results shown in Figure 5b,c indicate that the genes can display resistance to deactivation. The *sul3* gene had the lowest resistance due to the presence of possible dimers, including thymus pyrimidine (TT) dimers. TT dimers influence DNA copies and gene transcription by breaking the hydrogen bonds across double-stranded iDNA molecules, thus impeding DNA synthesis [40,43,44,45].

The HGT frequency was compared between MW and UV irradiation. It was observed that MW (0.008) detected a lower frequency of ARGs that were released from damaged *E. coli* 10667 (*sul*) into the environment recipient (forward transfer) than UV (0.014). Similarly, we explored the reverse transfer of RP4 plasmid. MW (0.003) displayed a higher frequency than UV (0.002) from the environment bacterium to the damaged *E. coli* 10667 (*sul*), as presented in Figure 5d. The high reverse transfer frequency under MW irradiation improved the cell membrane penetrability of the survival damaged *E. coli* 10667 (*sul*). Meanwhile, the total HGT frequency of UV irradiation was higher than that of MW irradiation.

In summary, the above findings indicated that MW is an effective airborne disinfection technology for *E. coli* and can control the transmission of AR. However, future research should consider the transmission of AR with MW disinfection for other microorganisms.

### 3.4. Survival of Concentrations of Various Strains at Different Irradiation Times

In Figure 6a, the survival concentrations under the MW irradiation based on absorbing material under different irradiation times in various strains are reported. Under UV irradiation, the concentrations of *E. coli* DH5α with RP4, strains of *E. coli* GMCC13373, and the concentration of strain *E. coli* CICC 10667 were obtained, as shown in Figure 6b. The result showed that *E. coli* GMCC13373 was highly resistant to MW, whereas *E. coli* CICC 10667 and *E. coli* DH5α with RP4 plasmid were easily inactivated and showed similar survival concentrations. By contrast, *E. coli* CICC 10667 presented high resistance to UV, whereas the other two strains showed similar results under UV treatment.

### 3.5. *Eenergy Efficiency per Order* (*EE/O) of Various Strains at Different Power Outputs under MW or UV Irradiation*

In Figure 7, the EE/O (kJ/m^3^) describes the electrical energy required to degrade various strains by one order of magnitude for the assessment of energy requirement (Equation (4)) [46].
(4)$$EE/O=\frac{P\times T}{1000\times V\times \mathrm{lg}(C_{0}/C_{t})}$$
where *EE/O* is the energy required to degrade airborne by one order of magnitude (kJ/m^3^), *P* shows the power of the device (Watt), *T* is the time of reaction (s), *V* is the volume of airborne (m^3^), *C_o_* is the initial concentration of airborne microbe before MW or UV irradiation, and *C_t_* is the concentration after treatment. The *EE/O* values after MW and UV treatments are shown in Figure 7. The present study showed that the energy consumption of MW was higher than that under UV irradiation.

### 3.6. Mechanism of HGT under MW or UV Applications

A comparison of MW and UV disinfection in HGT is depicted in Figure 8. The *E. coli* GMCC 13373 was used as the recipient in forward transfer, and *E. coli* DH5α was used as the donor in reverse transfer. Under MW irradiation, iARGs were released from the cell into the airborne environment and became a source for the spread of environmental eARGs. Forward transfer occurred when the recipient bacteria absorbed naked DNA from deceased donor bacteria in the environment and incorporated it into their chromosomes or transformed it into an autonomous extra-chromosomal replicon [47].

However, a plasmid DNA can be transferred through a channel by interacting with a donor-carrying plasmid and recipient bacteria through a pilus or pore in the reverse transfer process [48]. A double-stranded DNA (plasmid) molecule imitates self-sufficiently and reproduces genetic material, which is transported to a living cell and nucleus [49]. UV irradiation may damage DNA by penetrating a cell wall rather than producing free radicals. By contrast, the rate of reverse transfer under UV irradiation was significantly lower than that under MW irradiation. The above results showed a connection between treated *E. coli* 10667 (*sul*) and surrounding bacteria. The released ARGs disseminated through the forward and reverse transfer mechanisms of HGT in the environment.

## 4. Conclusions

The present study investigated the effects of MW or UV irradiation on ARBs and ARGs. The strains of *E. coli* CICC 10667 (5.5-log) and *E. coli* DH5α with RP4 plasmid (5.0-log) were more accessible for inactivation than a common strain of *E. coli* CGMCC 13373 (3.4-log) under MW irradiation. During UV disinfection, *E. coli* DH5α with the RP4 plasmid achieved a log inactivation value of 4.4-log, and *E. coli CGMCC* 13373 had 2.3-log. *E. coli* CICC 10667 had 2.1-log. *EE/O* was higher under MW irradiation than under UV irradiation.

The impacts of ARG deactivation and HGT by MW were compared with those under UV irradiation. The ARG inactivation rate (85.5%) under MW irradiation was higher than that under UV irradiation (48.2%). Similarly, MW significantly decreased the HGT frequency (0.008) of the released ARGs into the recipient (forward transfer) relative to UV (0.014). MW (0.003) was slightly higher when compared with UV (0.002), and the permeability of surviving damaged *E. coli* 10667 (*sul*) cells was increased and transferred with plasmid RP4 from the donor (reverse transfer). Moreover, the mechanisms of *E. coli* 10667 (*sul*) deactivation and injured DNA under MW and UV treatments were analyzed. In summary, MW treatment is a promising method for the airborne disinfection and control of ARGs.

## Figures and Tables

**Figure 1 ijerph-19-04332-f001:**
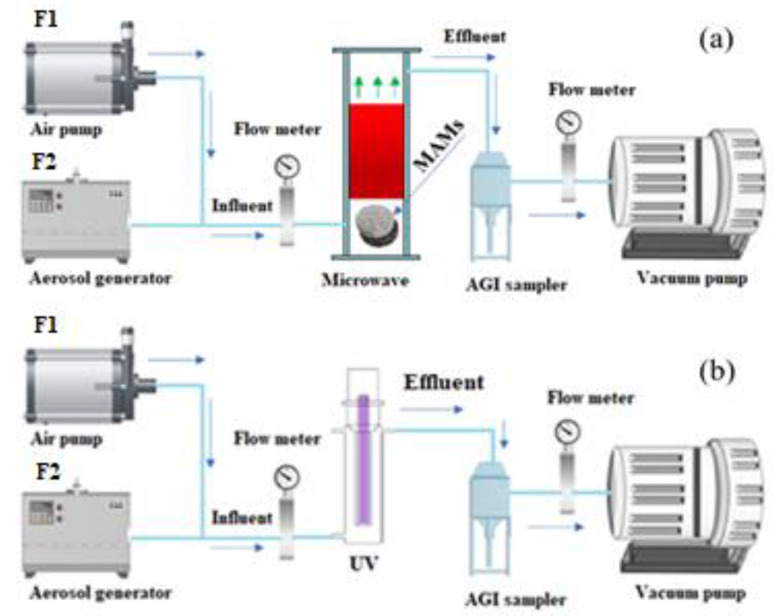
(**a**) Experimental setup for airborne ARB exposure to (**a**) MW and (**b**) UV irradiation MAMs: microwave absorbing materials.

**Figure 2 ijerph-19-04332-f002:**
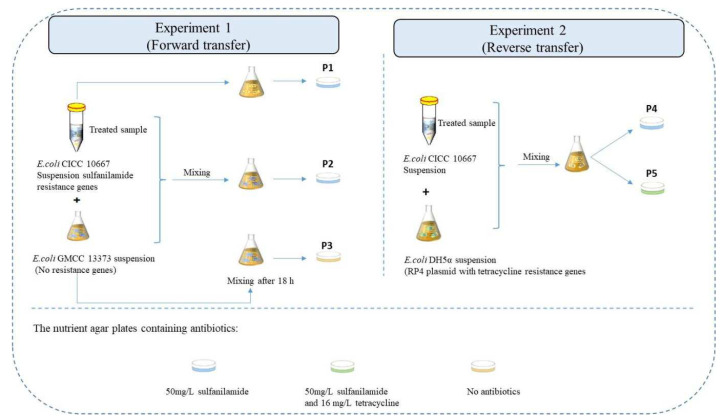
Process of the HGT experimental design. Experiment 1 (forward transfer): treated sample and *E. coli* GMCC 13373 as donor and recipient strains, respectively; Experiment 2 (reverse transfer): treated sample and *E. coli* DH5α as recipient and donor strains, respectively.

**Figure 3 ijerph-19-04332-f003:**
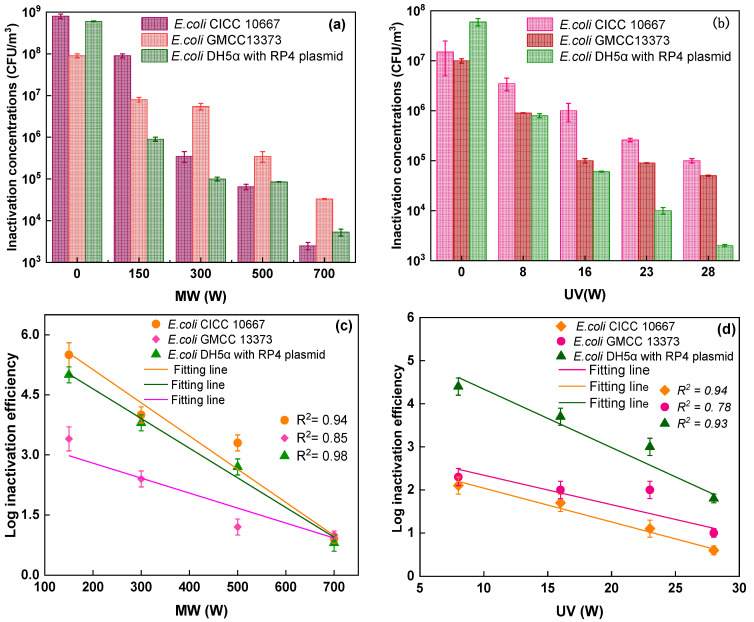
Inactivation performance of various strains of *E. coli* (CICC 10667), *E. coli* (GMCC13373), and *E. coli* DH5α with RP4 plasmid under MW irradiation based on MAMs at different power outputs (150, 300, 500, and 700 W), and UV irradiation at different power outputs (8, 16, 23, and 28 W) at 20 s irradiation time: (**a**) MW concentrations, (**b**) UV concentrations, (**c**) MW log efficiency, and (**d**) UV log efficiency.

**Figure 4 ijerph-19-04332-f004:**
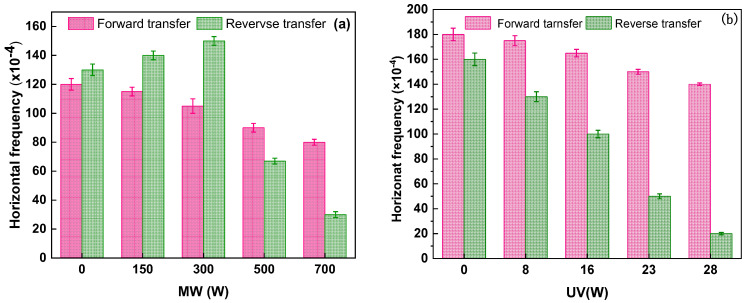
Effect of various power outputs on the forward and reverse transfer frequency under (**a**) MW or (**b**) UV irradiation. The concentration of bacteria was 10^8^ CFU/m^3^, the donor/recipient ratio was 1:1, and the reaction time was 16 h. The error line indicated the standard deviation of the three repeated tests.

**Figure 5 ijerph-19-04332-f005:**
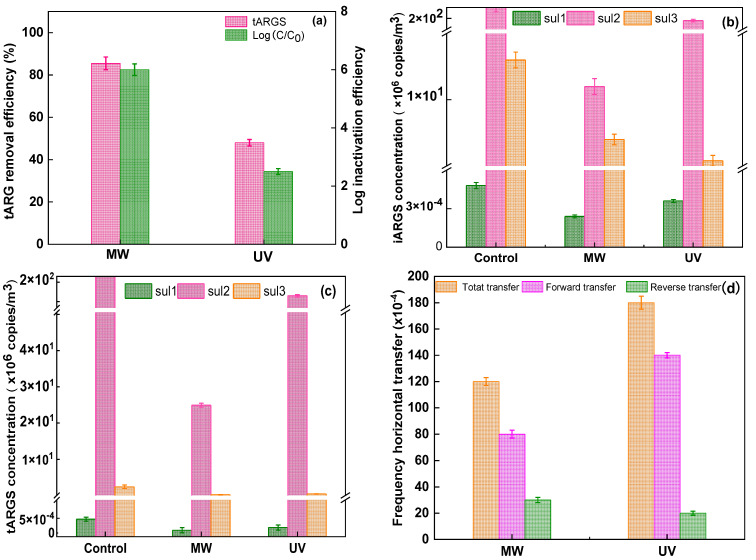
Comparison of MW and UV on (**a**) the removal of *E. coli* 10667 (*sul*) and tARGs, (**b**) individual iARGs concentrations, (**c**) tARG concentrations, and (**d**) effects of MW and UV irradiation on HGT frequency.

**Figure 6 ijerph-19-04332-f006:**
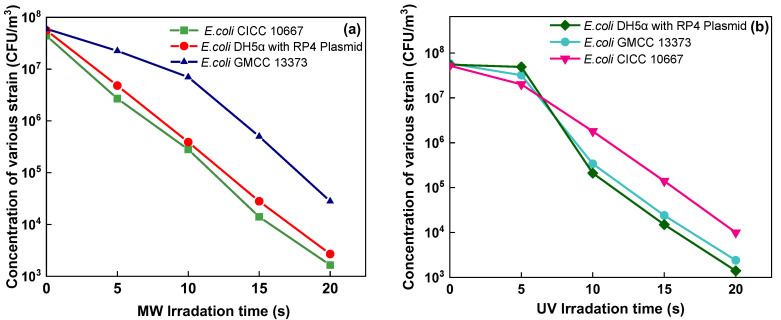
Concentration of various strains at different times of exposure to (**a**) MW and (**b**) UV.

**Figure 7 ijerph-19-04332-f007:**
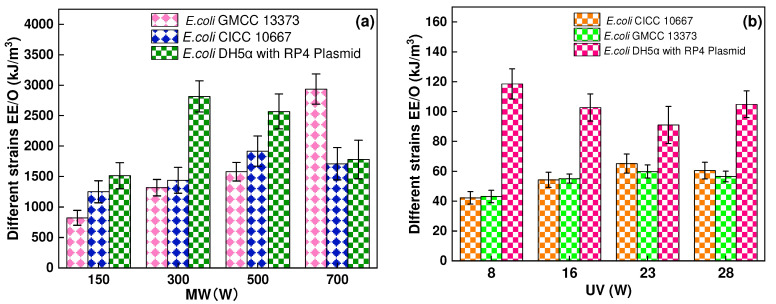
*EE/O* of various strains of *E. coli* (GMCC13373), *E. coli* (CICC10667), and *E. coli* DH5α with RP4 plasmid under (**a**) MW irradiation at different power outputs (150, 300, 500, and 700 W), (**b**) UV irradiation at various power outputs (8, 16, 23, and 28 W) at 20 s irradiation time.

**Figure 8 ijerph-19-04332-f008:**
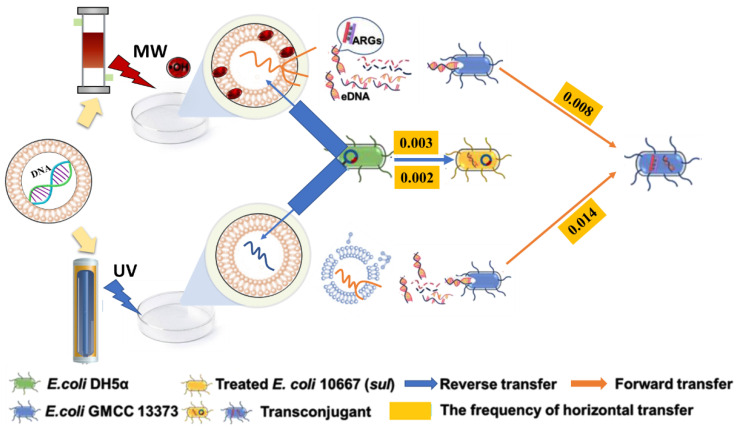
Mechanisms for HGT under MW or UV irradiation. Forward transfer (*E. coli* GMCC 13,373 as recipients) and reverse transfer (*E. coli* DH5α as donors).

## Data Availability

The study did not report any data.

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
