# Peer review of "Mechanism for Reducing the Horizontal Transfer Risk of the Airborne Antibiotic-Resistant Genes of Escherichia coli Species through Microwave or UV Irradiation"

_ijerph, 2022, doi:10.3390/ijerph19074332_

Round 1

Reviewer 1 Report

The research idea of this manuscript is to reduce the HGT among bacteria, focusing on antibiotic resistance genes. The subject is important, but the way the manuscript is written is difficult to follow and misleading because only a few strains of E. coli have been studied. It is not clear why these physical methods should specifically reduce HGT; these methods are generally detrimental for bacteria. According to all these, I can't support this manuscript for publication.

Author Response

We hope you will be satisfied with our answers and this revised manuscript. Thank you once again.

Reviewer 2 Report

 On account of the manuscript IJERPH-1618444, entitled “Mechanism for reducing the horizontal transfer risk of airborne antibiotic resistance genes across Escherichia coli species by microwave or UV irradiation” by Azhar Ali Laghari et al., the authors evaluated the inactivation of airborne E. coli 10667 (containing sul genes) and ARGs based on the microwave (MW) and ultraviloet (UV) irradiation. The topic is important to conduct environmental risk management of antimicrobial resistance in the water environment, and the authors got interesting results. The manuscript was well written and designed. After careful consideration, I made a decision that the manuscript is acceptable for publication in its present form.

Special remarks:

‧ The present manuscript evaluated the efficiency of inactivation for airborne E. coli 10667 (containing sul genes) and ARGs by microwave (MW) and ultraviloet (UV) irradiation, and further discussed the horizontal gene transfer (HGT) risk after treatment.

‧ The environmental risks of antimicrobial resistance in the water environmental have been concerned recently. However, few researchers have examined the impacts of airborne ARB deactivation on HGT risk. This aspect is considered to new view point and interesting.

‧ The present manuscript provided useful prospects to better understandings for the environmental management of antimicrobial resistance, and human health risk assessment as well.

‧ The interpretation of the evidence and arguments presented and conclusions are sufficient.

‧ The references cited relevant and up to date.

‧ The tables and/or figures are useful, necessary, and good quality.

Author Response

I am thankful for your kind concern. I am honored that you have accepted our manuscript thank you so much again. 

This manuscript is a resubmission of an earlier submission. The following is a list of the peer review reports and author responses from that submission.

Round 1

Reviewer 1 Report

It is an interesting study exploring the techniques for effectively eradicating the ARB and ARG and the associated HGT's. The microwave based platform appears promising and the comparative results (versus UV radiation) seems quite interesting. The results from this study pave the way for newer ways of controlling the HGT's, beyond that the mechanism of HGT was also established. A few minor corrections, 1) Please explain abbreviations the first time it appears in the text (for example MW for microwave). 2) Figure 4, legends are incomplete.

Author Response

Thank you for your professional opinions and detailed suggestions. We have revised the paper according to the comments of the reviewers. We would like to response the comments of the reviewers point to point and hope that our responses are clear and satisfy your requirements.  Pleased find my attachment. 

Reviewer 2 Report

The manuscript “Airborne mechanism for reducing the horizontal transfer risk 1 of antibiotic resistance genes across microbial species by micro- 2 wave or UV irradiation” investigate the inactivation of Antibiotic-resistant bacteria, the release, and removal of antibiotic-resistant genes by microwaves based technologies on absorbing materials (MAMs) or UV irradiation.

The idea is interesting, and overall the manuscript is well designed. There are no major drawbacks of the current manuscript. The English should be improved since the message is occasionally hard to understand and blunted.

Some minor aspects that should be improved in respect for this manuscript to be further considered:

-Please correct the references. There are zones in which the type does not respect the journal guidance.
- In several areas of the manuscript, there are some typos: - replace “induced” with “induce”, “shows aerosol generator” with “shows an aerosol generator”, “The  above  results  indicates” with “The  above  results  indicate” and correct “cylinderical-shaped”.

-Please rephrase some sentences which are hard to be understood, as e.g.: “ An effective method to make better use of microwave energy microwave absorbing materials (MAMS) can be used to overcome these problems.”

- Please explain “UV reported an active mechanism”

Author Response

(The authors gave the same response as above.)

Reviewer 3 Report

The manuscript of Ali Laghari aims to study the efficiency of horizontal gene transfer after treatment of bacteria with two physical methods: microwave and UV radiation. Although the subject is important and needs further investigation, this paper has in its present form many shortcomings which needs to be addressed.

  • The manuscript needs an extensive English correction; the present version is difficult to read, follow and at some places also not easy to understand.
  • The results have to be condensed in a nice story, based only on the results of analysis performed in this work, the mechanisms behind the effect of both techniques of conjugation efficiency as presented in Figure 8 is not supported by results.
  • The results have been obtained only with four strains representing one bacterial species. The results would be probably very different for other species, especially from Gram-positive bacteria, and this should be very clear from title, abstract and conclusions.
  • L128-L131: what kind of methods was used for DNA isolation
  • Where are encoded sulfanilamide resistance genes?
  • L152: What does represent conc. 16 mg/L? If plasmid, this is very high conc.
  • Both physical techniques applied in this work damage or inactivate the bacterial cells. The manuscript should present a clearer limitation/border of both techniques on cell damage and conjugation. This is probably very specific for species used in the study.
  • L406-L407: this statement needs explanation.
  • Conclusions: may be more focused on possible application of results obtained in this manuscript on diminishing HGS in environment.  

Author Response

(The authors gave the same response as above.)

Round 2

Reviewer 3 Report

The research idea of this manuscript is to reduce the HGT among bacteria, focusing on antibiotic resistance genes. The subject is important, but the way the manuscript is written is difficult to follow and as mentioned in my previous review title is misleading because only 4 strains have been studied and also since it is not clear why these physical methods should specifically reduce HGT; these methods are generally detrimental for bacteria. Besides, there are some statements that are contradictory, such as in L14, but also in the abstract in L15, and further on in-text. According to all these, I can't support this manuscript for publication.